# Complementary and Alternative Therapies in Oncology

**DOI:** 10.3390/ijerph19095071

**Published:** 2022-04-21

**Authors:** Agnieszka Dawczak-Dębicka, Joanna Kufel-Grabowska, Mikołaj Bartoszkiewicz, Adrian Perdyan, Jacek Jassem

**Affiliations:** 1Department of Clinical Oncology and Immunooncology, Greater Poland Cancer Center, 61-866 Poznań, Poland; adawczakdebicka@gmail.com; 2Department of Oncology, Poznan University of Medical Sciences, 61-701 Poznań, Poland; joannakufel@gmail.com; 3Department of Immunobiology, Poznan University of Medical Sciences, 61-701 Poznań, Poland; 4International Research Agenda 3P Medicine Laboratory, Medical University of Gdańsk, 80-210 Gdańsk, Poland; perdyan.adrian@gumed.edu.pl; 5Department of Oncology and Radiotherapy, Medical University of Gdańsk, 80-210 Gdańsk, Poland; jjassem@gumed.edu.pl

**Keywords:** cancer, alternative and complementary medicine, whole-body hyperthermia, chlorella, hemp, vitamin C, turmeric, ozone therapy, spirulina

## Abstract

Cancer is the second leading cause of death worldwide, after cardiovascular diseases. Increasing patients’ awareness and providing easier access to public information result in greater interest in alternative anticancer or unproven supportive therapies. Fear of cancer and limited trust in the treating physician are also important reasons leading patients to seek these methods. Trust and good communication are essential to achieving truthful collaboration between physicians and patients. Given the popularity of CAM, better knowledge about these alternative practices may help oncologists discuss this issue with their patients. This article objectively reviews the most common unconventional therapies used by cancer patients.

## 1. Introduction

Cancer is the second leading cause of death worldwide after cardiovascular diseases, and its incidence is growing. The efficacy of cancer treatment is increasing due to a better understanding of its biology and improvements in diagnostic and therapeutic methods. Active participation by patients in the diagnostic and therapeutic process may increase their compliance and well-being. However, greater patient awareness, more accessible public data, and determination often prompt them to seek unproven alternative therapies.

Complementary and alternative medicine (CAM), as opposed to evidence-based medicine (EBM), is not grounded in well-designed clinical studies, and thus may not be effective or may even harm patients. Complementary medicine is used in addition to standard medicine, whereas alternative medicine is used in lieu of standard methods.

Patients diagnosed with cancer are frequently confused due to the unpredictability of the situation, stress, and fear of the future of themselves and their families. The willingness to actively participate in the therapeutic process may prompt them to seek allegedly effective CAM options. Patients attempt these methods to increase treatment efficacy, alleviate treatment side effects, or improve their physical and mental condition. However, in many instances, patients replace main treatments with alternative methods, which may considerably worsen their prognosis.

The use of CAM in cancer patients has been consistently increasing [1]. For example, in a nationwide survey carried out in the Nepal, 32% of cancer patients reported using alternative therapies [2]. In another study of almost 1500 cancer survivors, 67% reported ever using CAM, and 43% had used CAM in the past year [3]. Alternative therapies are not subject to any formal regulations in Poland, and no public education programs address this issue. Consequently, patients often rely on knowledge from the Internet, which is frequently untrustworthy. The growing popularity and heterogeneity of CAM methods make them an important issue for patient–doctor relations in Poland and other Central European countries [4]. A recent study from Poland demonstrated that an astonishing number of CAM practices offered to manage multiple entities [5].

One of the reasons for seeking unconventional methods is the lack of time and understanding of medical staff. Cancer therapy requires a complete understanding of both parties and a truthful dialogue to ensure the safety and well-being of the patient. In addition, a sincere relationship with the treating physicians and their basic knowledge of alternative treatments may significantly influence patients’ decision-making process.

The increasing use of CAM by cancer patients constitutes a challenge for health care systems. Apart from social education, good communication between cancer patients and medical staff is crucial in managing this problem. This aim may be achieved by competence, understanding, patience, and adequate support for patients.

Health care professionals generally question the value of CAM and see no need to increase their expertise on this subject. However, having a basic knowledge of CAM may facilitate discussion with patients and influence their decisions.

This article summarizes the most common unconventional therapies used by cancer patients. It objectively presents their mechanisms and alleged anticancer effects. Given the popularity of CAM among patients, this knowledge may help clinicians discuss this issue with their patients.

## 2. Methods

In April 2021, we performed a literature search using PubMed, Scopus, and Google Scholar. We used the following search query: “cancer OR neoplasm AND (alternative medicine OR complementary medicine OR unconventional therapies OR whole-body hyperthermia OR chlorella OR beet juice OR carrot juice OR hemp OR propolis OR vitamin C OR turmeric OR ozone therapy OR spirulina OR acupuncture OR homeopathy)”. Additionally, articles were found in references and by Google search using similar terms. Both individual studies and overviews were included. Articles in languages other than English were excluded.

## 3. Results

Search results for particular therapies are presented below. The results of animal and in vitro studies, effects in humans, and treatment toxicity are presented in Table 1, Table 2 and Table 3, respectively.

### 3.1. Chlorella

Chlorella is a unicellular alga from the class of green algae that is increasingly being added to yogurts, juices, and smoothies in powder form. It is rich in protein; vitamins (particularly B vitamins); trace elements such as magnesium, potassium, iron, calcium, and zinc; fiber; and omega-3 fatty acids. The antioxidant and immunomodulatory properties of chlorella result from increasing the activity of NK cells and stimulating the production of interferon-γ, interleukin-12, and interleukin-1β. Ref. [62] Hot water extract of *Chlorella vulgaris* induces apoptosis and DNA damage in non-small-cell lung cancer cell lines [63]. Animal and in vitro studies have shown its antiproliferative effects on liver and colorectal cancer cells [6,7]. Lycopene isolated from chlorella inhibited the growth of prostate cancer cells in [64]. In an animal model, chlorella extract reduced bone marrow suppression caused by cisplatin [8]. Clinical data on chlorella treatment include only a small group of breast cancer patients [31]. According to a survey, chlorella extract decreased the severity of chronic weakness and dry skin in this group. The anticancer activity of chlorella has not been the subject of clinical trials.

### 3.2. Beet Juice, Carrot Juice

Many studies confirm the role of diet, mainly in cancer prevention. A diet including high amounts of lutein-rich vegetables such as spinach, broccoli, lettuce, tomatoes, oranges, carrots, and celery has been proven to reduce the risk of developing proliferative diseases. The inclusion of these foods in the diet can reduce the risk of colorectal cancer [65]. A citrus-rich diet reduces the risk of laryngeal cancer, and a diet high in fruits and vegetables reduces the risk of pancreatic cancer [65,66]. Consumption of carotenoid-rich foods inhibits DNA damage, and the betanin in beet juice induces apoptosis in breast cancer cells [67,68].

In vitro studies have shown that beet juice may increase the anticancer effect of doxorubicin [9]. A case report suggested that the combination of chlorambucil with beet juice and carrot juice is beneficial in a B-CLL leukemia patient [32]. Finally, the consumption of large amounts of carrot juice and beet juice was shown to reduce the anticancer effects of cisplatin [33].

### 3.3. Hemp

Hemp and hemp-derived cannabinoids (i.e., substances that act on cannabinoid receptors) are available for medical treatment in many countries. Individual preparations differ in delta-9-tetrahydrocannabinol and cannabidiol (CBD) content. In Poland, an aerosol preparation containing THC and CBD is registered to treat spasticity symptoms in patients with multiple sclerosis, but is not refunded. Until recently, hemp was also available in Poland in a dried form containing 19% THC and <1% CBD. This medicine does not have the characteristics of a medicinal product, and thus the specific indications for its use cannot be listed. The dried form is registered as a prescription ingredient.

Some studies have shown that cannabinoids might reduce the severity of nausea and vomiting associated with chemotherapy [34] and carry analgesic effects [35]. THC and CBD have alleviated acute and chronic pain in animal studies, but results in cancer patients are inconclusive [36]. A small study reported improved anxiety, mood, and well-being with cannabinoids in cancer patients [11]. However, no extensive clinical trials have been undertaken to confirm the analgesic effect of THC and CBD [69].

Anticancer effects of cannabinoids have only been tested on cancer cell lines. The administration of CBD increased the sensitivity of multiple glioma cells to chemotherapy [10]. Antiproliferative effects of THC and CBD have also been shown in breast, uterus, gastric, colorectal, pancreatic, lung adenocarcinoma, prostate cancer, and lymphoma cell lines [11].

The side effects of cannabinoids are primarily related to their stimulant and depressant effects on the CNS. Patients may experience confusion, impaired memory, drowsiness, and perceptual disturbances. Interestingly, unlike opioid receptors, cannabinoid receptors are not present in the brain’s respiratory center. Thus, in the case of cannabinoid overdose, there is no fear of respiratory depression [11]. However, THC and CBD affect receptors not only in the nervous system; they can cause tachycardia, hypotension, muscle relaxation, or impaired gastrointestinal motility [51]. The risk of cannabinoid addiction is lower than for tobacco, alcohol, or cocaine. Withdrawal symptoms such as irritability, restlessness, nausea, or insomnia have been less severe than those accompanying benzodiazepines or opiates, and usually resolve after a few days [11].

### 3.4. Propolis

Propolis is a resinous substance collected by bees from the buds and shoots of young trees and green plants. It is available as a dietary supplement in several forms, including pills, capsules, tablets, drops, syrups, ointments, sprays, powders, or liquids for skin application.

Studies on cell lines and animal models have shown a cytotoxic effect of propolis on breast, cervical, skin, gastric, prostate, and leukemia cancer cells, and a protective effect on the DNA of healthy cells [12,13,14,15]. A Polish study claimed that ethanol extract of propolis has cytotoxic activity on glioblastoma cell lines [70].

Due to its anti-inflammatory and antimicrobial effects, propolis extract was also used to treat complications after radiotherapy. A small study showed that propolis extract allowed the healing of radiation skin ulcers resistant to standard treatments [71]. Finally, in a study involving over 200 patients with breast cancer and head and neck cancer, propolis solution was shown to be effective and safe in preventing and treating oral mucositis caused by radiotherapy or chemotherapy [37]. However, larger clinical studies have not confirmed any beneficial effects in patients with radiation or postoperative ulcers and oral mucositis. So far, no study has shown the anticancer effects of propolis.

### 3.5. Vitamin C

Vitamin C is one of the most potent antioxidants. The first reports of its potential anticancer effects were published in the 1970s. An increasing number of institutions offer intravenous vitamin C infusions for cancer patients, advertising them as an adjunctive or anticancer modality.

High doses of vitamin C inhibit the growth of prostate, colon, and pancreatic cancer, as well as mesothelioma cell lines [16,17,18,19]. In a phase I study, the addition of high vitamin C doses to anticancer therapies (e.g., gemcitabine) was claimed to increase their effectiveness [19]. However, in other studies conducted on cell lines and in animal models, high doses of vitamin C reduced the effectiveness of chemotherapy [38,52,72]. In a small study, intravenous vitamin C infusions reduced chemotherapy-related symptoms such as fatigue, nausea, vomiting, or loss of appetite [38]. Currently, there is no evidence to confirm the beneficial effects of intravenous vitamin C in cancer patients, and its use may even reduce the effectiveness of treatment.

Due to the risk of hemolysis, intravenous vitamin C is contraindicated in patients with glucose-6-phosphate dehydrogenase deficiency [53], and even its oral form in these patients should be used with caution. High doses of vitamin C should not be administered in patients with a predisposition to kidney stones [38,52]. Vitamin C infusions administered shortly before chemotherapy may cause adverse interactions [52].

### 3.6. Turmeric

Turmeric is a spice originating from India and has been used in traditional Chinese and Ayurvedic medicine since ancient times. This compound has attracted great interest in recent decades because it contains bioactive curcuminoids (curcumin, demethoxycurcumin, and bisdemethoxycurcumin). Laboratory studies have shown its antioxidant and anti-inflammatory effects [20,21].

Turmeric was shown to increase the sensitivity of cancer cells to cisplatin, 5-fluorouracil, paclitaxel, and radiotherapy [21,22,23,24,73]. Some studies also demonstrated a chemoprotective effect of turmeric against the development of head and neck cancer and colorectal cancer [74,75].

The clinical data on turmeric are scarce. In patients with colorectal cancer, turmeric reduced weight loss and decreased serum inflammatory parameters [39]. A phase II trial involving 44 patients claimed that 30-day turmeric therapy might reduce tumor size [40]. Other studies involving small groups of patients with prostate and pancreatic cancer, that investigated this compound alone or combined with radiotherapy and chemotherapy, have been inconclusive. In patients with head and neck cancer and breast cancer, turmeric reduced radiation skin reactions [76,77].

No severe side effects of turmeric have been observed, but there are interactions with antiplatelet agents [54], doxorubicin, or tacrolimus [55]. Turmeric also affects the activity of cytochrome P450 enzymes [56].

### 3.7. Ozone Therapy

Ozone is one of the most potent disinfectants. Oxygen released from the ozone molecule damages the cell membrane of bacteria, leading to DNA injury and bacterial cell death [78].

In medicine, the most effective form of ozone therapy is autohemotransfusion. This procedure involves drawing approximately 100–150 mL of the patient’s venous blood, mixing it with ozone, and reinjecting it into the circulatory system.

Current data suggest that ozone therapy may increase the blood flow in tumor vessels, thus enhancing the efficacy of chemotherapy and radiotherapy. Studies on tumor cells and animal models have demonstrated the potentiation of fluorouracil by ozone in colorectal cancer [25]. Beneficial effects of ozone have also been claimed in the treatment of papillomavirus infection [79]. However, these effects have not been confirmed in clinical studies.

Ozone administered locally in ozonated water or oil, due to its antibacterial and anti-inflammatory properties, accelerated the healing of postoperative wounds and ulcers in patients with diabetes or local infections [41]. Some studies showed that ozone therapy might reduce nausea, vomiting, infections, hair loss, and weakness caused by anticancer treatment [80]. Anticancer effects of ozone in cancer patients have not been confirmed in the clinical setting.

Ozone therapy should not be used in pregnant women, patients with severe cardiovascular diseases (e.g., after myocardial infarction), hyperthyroidism, or thrombocytopenia [57]. In addition, special care should be taken in patients with asthma [58].

### 3.8. Spirulina

Spirulina is a protein-rich alga of the *Cyanophyta* division. It is a source of many nutrients, including D, K, and B vitamins, beta-carotene, and gamma-linolenic acid. This substance is available in pills, capsules, or powders, and can be added to smoothies, salads, or drinks. It has been attributed antibacterial and antiviral, immunomodulatory, anti-allergic, and anti-diabetic effects [81].

The assumption of spirulina’s potential anticancer effects is based on its antioxidant and protective properties on DNA structure [82]. In studies on cell lines of breast cancer and non-small-cell lung cancer, the phycocyanin contained in spirulina showed anti-angiogenic effects [26,82]. A study conducted in the 1990s in India showed the chemopreventive effect of spirulina in people chewing tobacco; approximately half of the patients who discontinued spirulina supplementation experienced the recurrence of oral leukoplakia [42].

### 3.9. Whole-Body Hyperthermia

Local hyperthermia combined with radiotherapy is an established treatment method for selected neoplasms. In turn, whole-body hyperthermia includes the planned and controlled increase of whole-body temperature over a specified time. In oncology, this technique is expected to kill cancer cells or inhibit their growth [83]. CAM uses whole-body hyperthermia alone or in combination with systemic chemotherapy to treat metastatic malignancies [84]. The increase in temperature may be achieved with thermal chambers or hot water mattresses [83,84,85,86]. Different therapeutic regimens have been used, including temperatures reaching 42 °C (extreme whole-body hyperthermia, the session lasts one hour), or temperatures between 39.5 and 41 °C (fever-like whole-body hyperthermia, the session may last longer, e.g., three hours).

There is currently no convincing evidence demonstrating the clinical efficacy of whole-body hyperthermia, given alone or in combination with other therapies, in cancer patients. Ongoing clinical trials investigate its combined use with radiotherapy and chemotherapy [43].

### 3.10. Acupuncture

Acupuncture is a part of traditional Chinese medicine. It involves the stimulation of specific points on the skin with metal needles. The method is based on the belief that vital energy (qi) flows through channels (meridians) in the human body; the role of acupuncture is to unblock the energy channels and improve the circulation of vital energy.

In cancer patients, acupuncture reduces tumor-related and postoperative pain [44]. The administration of morphine at “acupuncture points” allowed for longer pain control in [87]. Acupuncture also alleviates nausea, vomiting, and weakness and improves quality of life [45,46].

Acupuncture performed by an experienced therapist is relatively safe. However, it may be accompanied by pain or bleeding at the needle site, and some patients experience dizziness. Acupuncture should not be used in patients with hemophilia or those administered anticoagulants. Special care should be taken for patients with skin diseases [59].

### 3.11. Glutathione

Glutathione (GSH) is a major antioxidant produced in the human body. Its principal function is to protect biological systems from reactive oxygen species (ROS). The thiol group of glutathione reduces oxidizing species, whereas thiol itself is oxidized to a disulfide bond, resulting in the formation of glutathione disulfide (GSSG). In biological systems, the high ratio of GSH to GSSG (>100:1) is constantly preserved by GSSG reduction. The main obstacle with endogenous GSSG is its poor cell membrane permeability. Thus, current studies are focused on molecular modifications in order to increase its intracellular concentration.

Liposomal forms of GSSG in cells and mouse models inhibit the invasion ability of cancer cells and retard tumor proliferation [27]. Canfosfamide is a novel GSH analog consisting of a phosphorodiamidate group activated by glutathione-S-transferase P1-1, an enzyme overexpressed in many human cancers. When activated, it gains cytotoxicity towards cancer cells. Clinical studies of canfosfamide in combination with pegylated liposomal doxorubicin in patients with advanced ovarian cancer did not provide convincing evidence of its clinical efficacy [47,48,49].

### 3.12. Alpha-Lipoic Acid

Alpha-lipoic acid (ALA), also known as thioctic acid, is a short-chain fatty acid containing sulfur. It naturally occurs in green vegetables and meat and is synthesized endogenously in mitochondria [88]. ALA is a popular dietary supplement used in managing diabetes and obesity, as it reduces blood glucose levels, increases insulin sensitivity, and promotes weight loss. Moderate doses of ALA (orally—up to 2400 mg/day; intravenously—600 mg/day) did not induce adverse effects in [60].

ALA has gathered attention in cancer due to its antioxidant properties. It impacts various inflammatory signaling pathways [89]. In in vitro studies, ALA blocks TGF-beta, HMGB1, ERK1/2, and AKT signaling, leading to the inhibition of epithelial-mesenchymal transition and the sensitization of breast cancer cells to ionizing radiation [28,29]. It also inhibits the proliferation of gastric and non-small-cell lung cancer cells by blocking STAT3 binding to the MUC4 promoter region and by suppressing EGFR phosphorylation through Grb2 downregulation, respectively [90,91]. ALA was shown to suppress lung cancer growth in mice models through mTOR autophagy inhibition [92]. Finally, in preclinical studies, ALA alleviated chemotherapy and radiotherapy-induced toxicities [93,94]. No clinical studies have been conducted to verify the anticancer effects of ALA shown in preclinical models.

### 3.13. Chelation

The anticancer activity of iron chelators (ICs) has been attributed to their iron-depletion properties. Iron is an essential molecule for normal cells and cancer cells, and participates in DNA synthesis and ATP production. High intracellular iron concentrations may promote tumor initiation, growth, and metastasis. However, its depletion induces cell-cycle arrest and apoptosis in cell lines from various malignancies, including pancreas, breast, and colorectal cancer, as well as leukemia [93]. Despite the relatively safe profile of ICs, the main issues are poor pharmacokinetics, pharmacodynamics, and biodistribution, and lack of cancer affinity [61,95]. Hence, studies have focused on the chemical modifications of known ICs in order to overcome these difficulties.

One of the well-investigated ICs is deferoxamine, which is widely used for iron-overload diseases. Its anticancer properties were demonstrated on various cancer cell lines and in clinical trials in neuroblastoma, leukemia, and hepatocellular carcinoma [96,97,98]. However, its extremely short plasma half-life and poor cell membrane permeability significantly limit its therapeutic efficacy. Thus, ICs with high cell permeability, such as CPX (ciclopirox olamine) and Dp44mT, have been identified and designed [99,100].

The presumed anticancer effects of ICs include intracellular iron level decrease, induction of ROS, cell-cycle arrest, apoptosis, and tumor-suppressor gene expression. In addition, ICs influence specific pathways by Wnt/Beta-catenin, mTOR signaling inhibition, and alterations in the MAPK pathway. However, the exact molecular mechanisms underlying the iron-chelation-induced anticancer effect are not fully understood.

Anticancer activity has also been claimed for copper chelators. Recently, a pregnenolone acetate nucleus-based tetrazole derivative (ligand-L) was shown to promote cervical cancer cell death by generating intracellular ROS [101]. Another copper chelator, tetrathiomolybdate, is claimed to be an anticancer and anti-angiogenic remedy that acts throughout NF-kappaB pathway inhibition. Preclinical studies on head and neck and breast cancer suggested some activity of this compound, but the results of clinical trials were disappointing [50].

### 3.14. Homeopathy

Homeopathy is based on the assumption that highly diluted natural substances can cure diseases or alleviate their symptoms. However, the exact mechanisms of this phenomenon are not fully understood. Homeopathic remedies have been subject to numerous preclinical studies and randomized, double-blind, placebo-controlled clinical trials, which have been inconsistent [30]. In cancer patients, homeopathy combined with conventional treatment is claimed to increase their quality of life and overall survival [102,103,104]. A few studies reported even a complete regression of tumors with homeopathy when used as a single modality [105]. Other studies suggest that it alleviates dermatitis symptoms during adjuvant cancer therapy [106]. Anticancer properties of homeopathy were demonstrated on cell lines from human kidney, colon, brain, and breast cancer [107]. In mice inoculated with human prostate cancer or melanoma cells, homeopathy reduced tumor volume and inhibited the metastatic cascade, respectively [108]. These anecdotal reports have never been verified in large clinical studies, and the use of homeopathy in cancer patients is unjustified.

## 4. Discussion

We presented scientific data on the most common CAM methods used by cancer patients. We have purposely foregone our judgment of the presented CAM methods and have not evaluated the scientific value of particular publications. Nevertheless, the wealth of literature illustrates an enormous interest in this topic.

Many studies claim that there are potential benefits of some CAM therapies, but these conclusions are generally not firmly grounded in EBM. Notably, most of the cited articles include laboratory and animal studies. Clinical data comprise mainly anecdotal case reports or small series of patients, whereas well-designed prospective clinical trials are scarce. Hence, extrapolating such weak knowledge on CAM methods into clinical practice and encouraging their use in cancer care is unjustified and potentially dangerous.

The education of medical personnel and patients is essential in reducing the use of CAM in oncology. There is still insufficient information from medical personnel, which might significantly limit the use of CAM by patients. Including CAM teaching in the curriculum of medical studies would enhance the awareness of health professionals and would allow patients the opportunity for honest discussion and to obtain information based on medical knowledge.

We are aware of some limitations of our work. First, due to a large body of literature, we addressed only the most frequently used CAM methods. The format of this article is narrative, as the high diversity of data from the literature precluded a formal subject overview. We have done our best to objectively present studies that did and did not demonstrate the potential benefits of CAM therapies. However, given the limited enthusiasm of researchers to publish negative findings, publicly available data may be seriously biased.

## 5. Conclusions

The common use of CAM methods in cancer care remains a challenging clinical and emotional issue. Better education for health professionals and the development of a practitioner’s guide for patients may effectively address this problem. This should include EBM information about CAM, indicating therapies that could supplement conventional methods and those that should definitely be avoided. We believe this review will increase clinicians’ knowledge on CAM and improve communication with their patients.

## Figures and Tables

**Table 1 ijerph-19-05071-t001:** Animal and in vitro studies.

Method	Authors	Major Findings
**Chlorella**	Azamai et al. (2009) [6] Cha et al. (2008) [7]	Antiproliferative activity in liver and colorectal cancer cells.
	Lin et al. (2020) [8]	Reduced bone marrow suppression after cisplatin.
**Beet juice, carrot juice**	Govind et al. (2013) [9]	Improved anticancer effect of doxorubicin.
**Hemp**	Torres et al. (2011) [10]	Increasing chemosensitivity of glioblastoma cells.
	Gil et al. (2013) [11]	Antiproliferative effects in breast, uterus, gastric, colorectal, pancreatic and prostate cancer, lung adenocarcinoma and lymphoma cell lines.
**Propolis**	Khacha-ananda et al. [12] (2016), Seyhan et al. [13] (2019), Alizadeh et al. [14] (2015), López-Romero et al. (2018) [15]	Cytotoxic effect in breast, cervical, skin, gastric, and prostate cancer and leukemia cells; protective effect on the DNA of healthy cells.
**Vitamin C**	Chen et al. (2012) [16], Pathi et al. (2011) [17], Du et al. (2010) [18], Takemura et al. (2010) [19]	High doses inhibit growth of prostate, colon, and pancreatic cancer, as well as mesothelioma cell lines.
**Turmeric**	Streyczek et al. (2022), [20] Meng et al. (2013) [21]	Antioxidant and anti-inflammatory effects.
	Sreekanth et al. (2011) [22] Shakibaei et al. (2013), [23] Selvendiran et al. (2011) [24]	Increasing sensitivity of cancer cells to cisplatin, 5-fluorouracil, paclitaxel, and radiotherapy.
**Ozone therapy**	Vincenzo et al. (2017) [25]	Potentiation of fluorouracil effect in colorectal cancer.
**Spirulina** [26]	Ouhtit et al. (2014) [26]	Anti-angiogenic effect in breast and non-small-cell lung cancer cell lines.
**Whole-body hyperthermia**		No studies identified.
**Acupuncture**		No studies identified.
**Glutathione**	Sadhu et al. (2017) [27]	Inhibiting invasion of cancer cells and tumor proliferation in mouse models.
**Alpha-lipoic acid**	Li et al. (2021) [28], Choi et al. (2020) [29]	Blocking TGF-beta, HMGB1, ERK1/2, and AKT signaling, leading to the inhibition of epithelial-mesenchymal transition and radiosensitization of breast cancer cells.
**Chelation**	Eberhard et al. (2009), Yuan et al. (2004)	High-cell-permeability ICs such as CPX (ciclopirox olamine) and Dp44mT have been identified and designed.
**Homeopathy**	Frenkel et al. (2015) [30]	Inconsistent results of preclinical studies and randomized double-blind, placebo-controlled clinical trials.

**Table 2 ijerph-19-05071-t002:** Effects in humans.

Method	Authors	Major Findings
**Chlorella**	Noguchi et al. (2014) [31]	Decreased severity of chronic weakness and skin dryness.
**Beet juice, carrot juice**	Shakib et al. (2015) [32], Szefler et al. (2019) [33]	Beneficial effect in combination with chlorambucil in a patient with B-CLL leukemia. Reduced anticancer effects of cisplatin with consumption of large amounts.
**Hemp**	Pagano et al. (2022), [34] Johnson et al. (2010), [35] Johnson et al. (2013) [36]	Reduced severity of nausea and vomiting associated with chemotherapy; analgesic effects.
**Propolis**	Münstedt et al. (2019) [37]	Effective and safe in preventing and treating oral mucositis caused by radiotherapy or chemotherapy.
**Vitamin C**	Welsh et al. (2013) [38]	Reducing chemotherapy-related symptoms such as fatigue, nausea, vomiting, and loss of appetite.
**Turmeric**	He et al. (2011) [39], Carroll et al. (2011) [40]	Reducing weight loss and decreasing serum inflammatory parameters in patients with colorectal cancer; reducing tumor size.
**Ozone therapy**	Tirelli et al. (2018) [41]	Reducing nausea, vomiting, infections, hair loss, and weakness caused by anticancer treatment.
**Spirulina**	Mathew et al. (1995) [42]	Chemopreventive effect in people chewing tobacco; recurrence of oral leukoplakia with discontinued administration.
**Whole-body hyperthermia**	Lassche et al. (2019) [43]	No convincing evidence of clinical efficacy.
**Acupuncture**	Ashby et al. (2021) [44]	Reduction of tumor-related and postoperative pain.
	Zhang et al. (2021) [45], Soo et al. (2017) [46]	Alleviating nausea, vomiting, and weakness; improving quality of life.
**Glutathione**	Rose et al. (2007) [47], Vergote et al. (2009) [48], Kavanagh et al. (2010) [49]	No clinical effect in combination with pegylated liposomal doxorubicin in patients with advanced ovarian cancer.
**Alpha-lipoic acid**		No studies identified.
**Chelation**	Khan et al. (2009) [50]	No convincing evidence of clinical efficacy.
**Homeopathy**	Frenkel et al. (2015) [30]	Negative results of large clinical studies.

**Table 3 ijerph-19-05071-t003:** Side effects and contraindications.

Method	Authors	Major Findings
**Chlorella**	Noguchi et al. (2014) [31]	Constipation.
**Beet juice, carrot juice**		No adverse events.
**Hemp**	Cohen et al. (2019) [51]	Tachycardia, hypotension, anxiety, euphoria, muscle relaxation, impaired cognitive functions.
	Gil et al. (2013) [11]	Withdrawal symptoms: irritability, restlessness, nausea, insomnia.
**Propolis**		No adverse events.
**Vitamin C**	Klimant et al. (2018) [52]	Infused shortly before chemotherapy may cause adverse interactions (e.g., red blood cell hemolytic crisis).
	Lo et al. (2020) [53]	Risk of hemolysis with intravenous infusion; contraindicated in patients with glucose-6-phosphate dehydrogenase deficiency.
**Turmeric**	Jantan et al. (2008) [54] Abushammala et al. (2021) [55] Zhang et al. (2008) [56]	Interactions with antiplatelet agents, doxorubicin, or tacrolimus, affecting the activity of cytochrome P450 enzymes.
**Ozone therapy**	Bocci et al. (2010) [57] Pepper et al. (2020) [58]	Contraindicated in pregnant women, patients with severe cardiovascular diseases (e.g., after myocardial infarction), hyperthyroidism, and thrombocytopenia; special care in patients with asthma.
**Spirulina**	Mathew et al. (1995) [42]	No adverse events.
**Whole-body hyperthermia**	Lassche et al. (2019) [43]	Grade 3/4 ventricular cardiac arrhythmias, dermal complications, nephrotoxicity.
**Acupuncture**	Höxtermann et al. (2022) [59]	Contraindicated in patients with hemophilia or administered anticoagulants. Special care in patients with skin diseases.
**Glutathione**	Kavanagh et al. (2010) [49]	Nausea, vomiting, rash, fatigue, neutropenia, leukopenia, anemia.
**Alpha-lipoic acid**	El Barky et al. (2017) [60]	No adverse effects with moderate doses.
**Chelation**	Komoto et al. (2021) [61]	No severe adverse events.
**Homeopathy**		No adverse events reported.

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
