# Peer review of "Complementary and Alternative Therapies in Oncology"

_ijerph, 2022, doi:10.3390/ijerph19095071_

Round 1

Reviewer 1 Report

  1. Some of the most common unconventional therapies used by cancer 
    patients are missing in the manuscript.
  2. There are no figures in the manuscript. The authors should add some diagrams (atleast 3-4) to make the article interesting for the readers.
  3. The discussion section is very small. The authors should elaborate on it.

Author Response

Dear Reviewer,

Thank you for each of your comments on our manuscript.

We added 3 figures in the manuscript. We hope that these figures will be more interesting for readers and improves our manuscript.

We enlarged discussion.

Reviewer 2 Report

The Authors make a comprehensive narrative review of Complementary and Alternative Therapies in Oncology. This is  a very important topic. Anyway I would recommend the Authors to better clarify complementary and alternative medicines definitions which sometimes have been put together. Tha majority of the arguments in their work deal with complementary and not alternative. So a clear distintion is warranted.

Maybe some more comments in the conclusions subchapter may be useful regarding the use of CAM.

Author Response

Dear Reviewer, 

Thank you for each of your comments. We clarified Complementary and Alternative methods. We added comments to conclusion. 

Reviewer 3 Report

Dear Authors,

I’m writing here to submit a revision for an article that I have recently received. The article “Complementary and Alternative Therapies in Oncology ” by A. Dawczak-DÄ™bicka  et al., aimed on characterization of natural compounds and alternative anti-cancer therapies. The authors propose that alternative anti-cancer or unproven supportive therapies create difficulties in diagnostic and treatment of patients with malignancies.  The current work attempted to summarize the most "popular"substances used in complementary and alternative medicine (CAM). 

Here some suggestions:

  1. I would recommend adding a table before the conclusion section to summarize side effects and interferenceof CAMs with traditional anti-cancer agents.
  2. As the conclusion part it would be great to have an idea about works that aimed to educate patients and medical personnel.

Author Response

Dear Reviewer,

Thank you for each of your comments on our manuscript. We added 3 figures in the manuscript. We hope that these figures will be more interesting for readers and improves our manuscript.

Reviewer 4 Report

The work offers a good description on the subject.
However, it does not provide an overview.

1)To make the description easily understandable, a descriptive table would be needed.
2) In the table you should enter the various elements of CAM, biological effects, in vivo or in vitro studies 3) Update the bibliography 4)

Author Response

Dear reviewer,

Thank you for each of your comments on our manuscript. We added 3 figures in the manuscript. We hope that these figures will be more interesting for readers and improves our manuscript.

Round 2

Reviewer 2 Report

Thank you

I think the Authors have clarified the point

Reviewer 3 Report

The major suggestions and comments were adressed.

Reviewer 4 Report

The review improved the paper and was done correctly